# A Speech Recognition Method Based on Domain-Specific Datasets and Confidence Decision Networks

**DOI:** 10.3390/s23136036

**Published:** 2023-06-29

**Authors:** Zhe Dong, Qianqian Ding, Weifeng Zhai, Meng Zhou

**Affiliations:** School of Electrical and Control Engineering, North China University of Technology, Beijing 100041, China; lilac77qian@163.com (Q.D.); zhaiwf@ncut.edu.cn (W.Z.); zhoumeng@ncut.edu.cn (M.Z.)

**Keywords:** speech networks, confidence decision making, domain specific, CTC

## Abstract

This paper proposes a speech recognition method based on a domain-specific language speech network (DSL-Net) and a confidence decision network (CD-Net). The method involves automatically training a domain-specific dataset, using pre-trained model parameters for migration learning, and obtaining a domain-specific speech model. Importance sampling weights were set for the trained domain-specific speech model, which was then integrated with the trained speech model from the benchmark dataset. This integration automatically expands the lexical content of the model to accommodate the input speech based on the lexicon and language model. The adaptation attempts to address the issue of out-of-vocabulary words that are likely to arise in most realistic scenarios and utilizes external knowledge sources to extend the existing language model. By doing so, the approach enhances the adaptability of the language model in new domains or scenarios and improves the prediction accuracy of the model. For domain-specific vocabulary recognition, a deep fully convolutional neural network (DFCNN) and a candidate temporal classification (CTC)-based approach were employed to achieve effective recognition of domain-specific vocabulary. Furthermore, a confidence-based classifier was added to enhance the accuracy and robustness of the overall approach. In the experiments, the method was tested on a proprietary domain audio dataset and compared with an automatic speech recognition (ASR) system trained on a large-scale dataset. Based on experimental verification, the model achieved an accuracy improvement from 82% to 91% in the medical domain. The inclusion of domain-specific datasets resulted in a 5% to 7% enhancement over the baseline, while the introduction of model confidence further improved the baseline by 3% to 5%. These findings demonstrate the significance of incorporating domain-specific datasets and model confidence in advancing speech recognition technology.

## 1. Introduction

Speech recognition technology is gaining increasing importance as an advanced technology for converting speech signals into text. With the expanding range of applications, it has become a key tool for enhancing human productivity and efficiency by replacing traditional typing and mouse operations with spoken input. This enables individuals to perform various tasks, including office work and learning, more efficiently [1]. Given this context, speech recognition technology is receiving growing attention and significance, with both academia and industry actively promoting its development and improvement.

In recent years, there have been significant advancements in ASR, with deep learning techniques emerging as a key development. Deep learning models such as convolutional neural networks (CNN) [2] and recurrent neural networks (RNN) [3] have become prominent methods for speech recognition. Additionally, end-to-end [4] speech recognition has simplified the architecture of ASR systems, improving their performance and efficiency. Machine learning methods, such as transfer learning [5], have also been employed for phoneme or word selection [6], especially in noisy and complex environments. These technological advancements have expanded the applications of speech recognition. With the advancement of technology and the growing need for human-computer interaction [7], speech recognition has found widespread use across various fields. Domain-specific speech recognition, for instance, converts speech signals into text or commands, facilitating natural, efficient, and convenient human-computer interaction [8]. Keyword recognition methods [9] are effective for phrase recognition in domain-specific contexts, but they require prior identification of keywords and are not suitable for handling unknown speech content. Unlike traditional Hidden Markov Model (HMM)-based speech recognition methods that only consider short-term transition relationships between consecutive frames, our study proposes a novel approach called DFCNN+CTC for speech recognition. By incorporating Deep Fully Convolutional Neural Networks (DFCNN) with the Connectionist Temporal Classification (CTC) framework, our method captures and leverages longer temporal dependencies, overcoming the limitations of HMM-based methods. Domain-specific speech recognition techniques [10] aim to develop specialized speech recognition models for specific professional domains, such as medical [11], legal, and financial fields. In domain-specific speech recognition, the recognition performance can be enhanced through the use of custom language models [12] or domain-adaptive methods [13,14]. These approaches employ techniques such as data augmentation [15], domain-adaptive training, and domain-specific model fine-tuning. Custom language models can be trained using a corpus of domain-specific vocabulary or adjusted by combining a general-purpose language model with domain-specific vocabulary. This helps reduce domain differences and improves recognition accuracy. In domain-specific contexts, the terms and expressions used in speech inputs differ from those in general-purpose speech recognition techniques. This disparity poses challenges for accurately recognizing domain-specific content. To address this issue, this paper proposes a method for creating a speech network using domain-specific datasets.

In this paper, we propose a joint modeling [16] approach that leverages acoustic and language models to collaborate with each other in parsing and translating the speech signal, leading to more accurate recognition results. The main contributions of this work can be summarized as follows:(1)We propose an acoustic modeling approach that combines the speech spectrogram, DFCNN, and CTC. This approach utilizes the rich information provided by the speech spectrogram, the powerful feature extraction capability of DFCNN, and the sequence modeling capabilities of CTC without alignment. By handling speech signals of varying lengths, we achieved improved speech recognition results;(2)To address unfamiliar words in new domains, we present a comprehensive system based on N-gram technology and the construction of domain-specific datasets incorporated into language models. This enables speech recognition in new domains;(3)To optimize the model, we propose a speech confidence-based determination method that dynamically adjusts the use of the language model, thereby enhancing the accuracy of the speech recognition model;(4)We designed experimental comparisons using different domain datasets to verify the effectiveness of the proposed method in addressing domain-specific speech recognition. These experiments follow a step-by-step incremental model experimental approach.

## 2. Related Works

In 2017, Valin et al. [17] proposed a method for speech enhancement using neural networks. They employed a self-encoder-like neural network to learn the representation of speech signals, resulting in significant performance gains as demonstrated on multiple datasets. In 2018, Afouras [18] developed a Transformer-based sequence-to-sequence audiovisual speech recognition model. This model utilized pre-computed visual features and audio Log-Mel filter features as input, achieving state-of-the-art performance at that time. This advancement contributed to the rapid development in the field of audiovisual speech recognition. In 2019, Zhuang et al. [19] proposed an end-to-end speech recognition method based on deep convolutional neural networks (DCNN). This approach involved performing convolutional operations and then predicting the output text using a multilayer perceptron. Testing on multiple datasets demonstrated its effectiveness in speech recognition tasks. In 2021, Hinton et al. [20] proposed a method for acoustic modeling using a self-supervised learning approach. They employed a self-encoder that predicted intermediate features of speech to train the acoustic model. This allowed the acoustic model to better learn the representation of the speech signal, resulting in improved accuracy of speech recognition. When tested on multiple datasets, this approach achieved significant performance improvements.

Many previous scholars in domain-specific research on speech recognition techniques have made valuable contributions, providing important lessons and methods. For example, Vergin and Rivarol [21] achieved the recognition of monosyllabic words in continuous speech by preprocessing the speech signal and performing acoustic feature extraction. This work is considered one of the earliest applications of audio feature extraction methods and laid the foundation for later speech recognition research. Alejandro et al. [22] proposed a dynamic term discovery strategy that extends the functionality of language models within the system by incorporating knowledge from external resources. Chiranjibi Sitaula and Lachlan Burne et al. [23,24] proposed two methods for detecting bowel sounds in infants: one based on CNN and Laplace HSMM, and the other based on a bilinear fusion ensemble model. The proposed Laplace HSMM strategy can be introduced as a general modification, while the bilinear fusion ensemble model enables different levels of granularity in learning models from the data, leading to more accurate audio detection. Zaria Imran et al. [25] proposed a deep learning-based classification approach using long and short-term memory, along with a feature-based classifier, for performing heart sound classification.

## 3. Modeling Methods and Related Technologies

### 3.1. Overall ASR Model Architecture

In this paper, we propose a speech recognition method based on the architecture of DSL-Net and CD-Net. As shown in Figure 1, Model 1, in this case, first utilized a domain-specific public dataset and automatically trained the dataset through DSL-Net. Then, the domain-specific speech model was trained on the pre-trained model parameters through MODEL 2 migration learning. Based on this, an integrated learning approach was used to combine the domain-specific speech models with the speech models trained from the benchmark dataset, thereby improving the recognition accuracy. The method adapts to the input speech by automatically expanding the lexical content in the ASR model and dynamically adjusting the use of the language model based on speech confidence to enhance the accuracy of the speech recognition model. Due to the complexity of the speech signal and the multifaceted influence of the language model, optimizing whether the trained language model should be activated in certain situations becomes crucial. To address this, we introduced a speech confidence-based decision layer to the last layer of the convolutional neural network. Using the two-channel confidence as input, the decision network is obtained through model training, and the decision layer determines whether the current audio utilizes the trained language model or not. Additionally, the method automatically increases the lexicon content of the ASR model using a lexicon and language model and employs accuracy as an evaluation metric to enhance speech recognition accuracy by optimizing the usage of the language model.

In this paper, we utilized the framework of DSL-Net and CD-Net along with domain-specific datasets, pre-trained model parameters, and integrated learning methods. This enables us to achieve accurate speech recognition and automatically expand the lexical content in the ASR model. Additionally, we further enhanced the model’s accuracy through confidence networks. Our proposed approach brings significant improvements in speech recognition, particularly in new scenarios and domains.

### 3.2. Speech Recognition Based on Speech Spectrograms

In this paper, an acoustic modeling approach based on DFCNN and CTC is proposed. DFCNN is an innovative speech recognition framework proposed by KDDI in 2016, which has gained attention in current acoustic modeling research. In this paper, the DFCNN framework is adopted and combined with the CTC technique to model whole-sentence speech. This model design allows the output units to directly correspond to the final recognition results, resulting in more accurate and precise speech recognition. The process of constructing the acoustic model in a speech recognition system involves several steps. Firstly, the time-domain speech signal is transformed into a speech spectrogram through operations such as framing, windowing, Fourier transform, and logarithmic scaling. Figure 2 illustrates the process of converting audio into speech spectrogram feature sequences. In the speech spectrogram, the x-axis represents time, the y-axis represents frequency, and the intensity of the colors reflects the amplitude, with brighter colors indicating higher amplitude and darker colors indicating lower amplitude. The speech spectrogram captures the energy distribution information across different frequency bands, which serves as a foundation for subsequent feature extraction and model development.

In order to automatically extract speech spectrogram features, this paper utilized the DFCNN framework to construct a speech recognition model. This framework effectively captures the spectral information of speech signals and performs recognition using a model-based approach. However, the DFCNN framework is relatively less innovative in terms of innovation, and its underlying logic is relatively fixed. Future research can explore the reconstruction of the underlying logic of the DFCNN framework to enhance its innovativeness and performance. The DFCNN structure can be seen as transforming the speech signal into a two-dimensional image, where time and frequency correspond to the two dimensions. In the network, each frame of the audio signal underwent Fourier transformation to create a time–frequency map, which was then convolved using a 3 × 3 convolution kernel. The network took a four-dimensional tensor as input with a shape of (batch_size, 1600, 200, 1), where the last dimension 1 indicates that the audio signal was monophonic. The ReLU function was used as the activation function for each neuron. The fully connected layer parameters were updated using the back-propagation algorithm, and the output loss was calculated and propagated back through the network. By combining the DFCNN network with CTC, the complete acoustic model was able to handle speech signals of varying lengths and automatically extract high-level features. This combination significantly improved the accuracy and robustness of the acoustic model, as illustrated in Figure 3 below. In speech recognition tasks, speech data underwent preprocessing to be converted into the form of a speech spectrogram. This is because the speech spectrogram effectively characterizes the speech signal. These operations allow for the automatic learning and extraction of high-level features in the speech signal, such as energy and frequency variations in the audio spectrogram.

Next, the CTC model took the output phoneme sequence from the DFCNN network as the input sequence and established a mapping relationship to produce the corresponding text sequence. The CTC model is designed to handle speech signals of varying lengths by incorporating a blank character and employing the B operations. Finally, using the maximum likelihood criterion, the model calculated the conditional probability and mapped the input speech spectrogram to the corresponding text output, thereby achieving the function of speech recognition.

When constructing the acoustic model, it was observed that considering the performance of individual phonemes alone was insufficient in terms of synchronic articulation. Therefore, the model initially employed the three-phoneme-based Gaussian Mixture Model–Hidden Markov Model (GMM-HMM) as the language model. The preceding and following phonemes of the current phoneme were considered as a unit, referred to as the triphone, and represented in the form of A − B + C. Here, B represents the current phoneme, A represents the preceding phoneme of B, and C represents the following phoneme of B. However, the three-phoneme model can encounter the issue of sparse data due to the larger number of modeling units involved. To address this problem and enhance model accuracy, a GMM-HMM model with a decision tree was introduced.

### 3.3. Domain-Specific Dataset Speech Network

(1)Construction of the domain-specific dataset

To improve the performance of acoustic models, this paper constructs a domain-specific dataset using a novel and innovative approach. Unlike previous studies that rely on generic datasets, the approach involves collecting and curating a large corpus of domain-specific audio samples. The construction of this dataset involved targeted data collection from relevant sources, rigorous quality control measures, and careful annotation to ensure the inclusion of domain-specific representative examples. This new approach to constructing domain-specific datasets allows our model to capture and exploit the unique features and complexity of the target domain, thereby improving recognition accuracy and performance.

In this paper, we constructed an ASR system using the aforementioned modeling approach based on the speech spectrogram and CTC sequences. Additionally, we created a domain-specific dataset by calculating the edit distance of text sequences. The edit distance formula (1) considers the words of insertion (I), deletion (D), and substitution (S), where $m=\left|ref\right|,n=|hyp|$ represent the lengths of the reference and hypothesis text sequences, respectively. The terms ref,hyp denote the annotated text and the speech recognition results. To establish the backtracking method, the priority order of substitution > insert > delete was utilized. By calculating the minimum value of the total number of operations and considering the sequence and number of different operations, the optimization process of the decision process was solved using a typical dynamic programming algorithm.
(1)$$D_{i,j}=min\left\{\begin{array}{ccc} D_{i-1,j-1} & +0 & if\;u_{i}=v_{j}\\ D_{i-1,j-1} & +1 & \left(Replacement\right)\\ D_{i,j-1} & +1 & \left(Insertion\right)\\ D_{i-1,j} & +1 & \left(Deletion\right) \end{array}\right.$$

In the processing of audio datasets, after experimental validation, the energy-based speech segmentation algorithm will be used for such segmentation cases. This algorithm is combined with the lexical results of the edit distance algorithm to extract the corresponding audio datasets from the original audio data. To enhance the energy characteristics of the speech signal, preprocessing steps including filtering, denoising, and normalization are performed. The speech signal is then segmented into overlapping short-time frames, typically ranging from 10 to 30 ms. Through several experiments, a suitable threshold value is determined, which depends on the fluctuation range of the audio signal. Setting the threshold value above the average range of the acoustic wave proves to be the optimal parameter. Finally, the sound events within the threshold interval are combined to form longer speech segments, resulting in more accurate and stable domain-specific datasets.

(2)Language modeling

Speech recognition techniques heavily rely on language models to recognize words and language rules present in audio signals. However, the limited scope of these language models hinders their ability to encompass all words and language rules. Consequently, unknown words and language rules often go unrecognized during the speech recognition process. To address this issue, the edit distance algorithm can be employed to identify unknown words within the speech recognition system and dynamically add them to the lexicon of the ASR system. In this scenario, the N-gram technique proves valuable in leveraging word frequency information within the dataset to improve the identification of unknown words. By utilizing N-gram technology, a language model is generated to calculate the probability of a linguistic expression based on linguistic rules and word frequency information. This model then predicts word sequences in the audio signal and matches them with words in the dictionary during speech recognition. Essentially, N-gram technology enables the dynamic scaling of the language model to accommodate diverse linguistic and cultural environments. By training and optimizing large-scale datasets, N-gram technology significantly enhances the accuracy and coverage of language models, thereby bolstering the reliability of speech recognition systems. Furthermore, N-gram technology facilitates the dynamic addition of words, as well as the training and optimization of speech recognition systems. Through the use of different datasets, various dynamic words can be generated to suit specific linguistic and cultural contexts.

(3)Domain-specific speech recognition model building

For speech recognition tasks, migration learning of small datasets plays a crucial role, mainly due to the high cost associated with acquiring such datasets. In this study, a migration learning approach is employed to leverage an existing trained model on a large-scale dataset as the base model and enhance its performance by fine-tuning the model parameters using a small dataset. By optimizing Equation (2) to fine-tune the model parameters, a new fully-connected layer was added on top of the base model’s feature extraction layer. This additional layer was trained using a small dataset to update the weights of the fully-connected layer. This approach not only enhanced the model’s accuracy on small datasets but also significantly reduced the time and computational resources required for model training. Consequently, the combination of generic features learned from the base model and the specific features derived from small datasets contributes to the improved performance of the new model on small data sets.
(2)$$\theta =\arg\min\frac{1}{n}\sum_{i=1}^{n}L(f_{\theta }\left(x_{i}\right),y_{i})+\lambda R(\theta ,\theta_{base})$$

A new model parameter θ was obtained by fine-tuning the pre-trained model $\theta_{base}$ to address the speech recognition problem for small datasets. The loss function L was utilized to measure the error between the predicted value $f_{\theta }\left(x_{i}\right)$ and the true label $y_{i}$ Additionally, a regularization term was employed to constrain the size of the model parameter θ in order to prevent model overfitting.

To ensure full migration of the domain-specific dataset in the ASR system, the trained ASR system was used as a pre-trained model. The pre-trained model comprised a multilayer network with 32 convolutional kernels of size 3 × 3 for the first convolutional layer, 64 convolutional kernels of size 3 × 3 for the second convolutional layer, and 128 convolutional kernels of size 3 × 3 for the third convolutional layer. During the migration process, the convolutional and pooling layers in the neural network were frozen, and two MaxPooling2D layers were utilized to downscale the features. The number of input features of the fully connected layer in the ASR system was simplified. The optimization algorithm used for the model was Adam, with the following parameter settings: $lr=0.005,\;beta1=0.85,\;beta_{2}=0.9,\;decay=0.0,\;epsilon=10^{-8}$. The loss function is minimized by improving the training method, thereby adjusting the updated weights and bias parameters of the model.

Finally, the ASR system was weighted to integrate learning with the speech models trained on the aforementioned domain-specific dataset. This weight reflects the contribution of each model in terms of prediction, and is determined as follows:(3)model3=a1×odel1+a2×model2
where, as shown in Equation (3) above, m_speech1 denotes the ASR system, m_speech2 denotes the speech model trained on the domain-specific dataset, “a1” and “a2” are the weight coefficients of “m_speech1” and “m_speech2”, respectively. These coefficients are determined by evaluating “m_speech1” and “m_speech2” on the validation set and measuring them using an accuracy metric. To ensure that the sum of the weight coefficients is 1, we use the model performance as a normalization factor. Additionally, m_speech3 represents the synthesized acoustic model. The combined model was able to fully capture the speech features of the domain-specific dataset.

### 3.4. Confidence Decision Model

Due to the complexity of speech signals and the multifaceted impact of language model modeling, further optimization is required to determine whether to apply the trained language model in specific scenarios. In this paper, we propose an innovative approach to dynamically adjust the use of language models through the determination of two-channel speech confidence. This novel confidence network can improve the accuracy of the speech recognition model by adaptively controlling the language model based on the reliability and confidence of the speech signal. Figure 4 illustrates the core concept of this approach, which involves adding a speech confidence-based decision layer to the final layer of the domain-specific dataset speech network. The input x_label represents the channel with higher dual-channel confidence, while the user group speech dataset is converted into an audio feature vector as input x1. By training the decision network using convolutional techniques with x_label and x1 as inputs, a decision model is obtained. The decision layer determines whether to employ the trained language model in a given speech environment, thus improving the speech recognition accuracy in diverse new scenarios and domains.

In order to enhance the accuracy of speech recognition, this paper developed a convolutional neural network model using the Keras library. The model architecture comprised three convolutional layers, two pooling layers, a Flatten layer, and two fully connected layers. A sigmoid function was applied for binary classification. The model was trained using the Adam optimizer with binary cross-entropy as the loss function, and accuracy was used as the evaluation metric. By analyzing the two-channel confidence of the speech signal, the model generated an output value that determined whether to utilize the trained language model. This process improved the accuracy of speech recognition. A threshold, denoted as σ, is set within the neuron to regulate the decision network’s division range. If the model output exceeded σ, the trained speech model was deactivated, and the Model 1 speech network was utilized. Conversely, if the output was below σ, the Model 2 speech network was employed for audio recognition.

The network comprised three convolutional layers and the tensor size of the first convolutional layer varied, as depicted in Figure 5 below. Each convolutional layer was designed to extract different features from the input audio signal by utilizing different filter sizes, kernel sizes, and activation functions. To ensure output value normalization and enhance the training process, a batch normalization layer was applied to normalize the output of each convolutional layer. In these convolutional layers, the typical convolutional kernel size was (3, 3), and the extracted feature values were either 32 or 64. Additionally, the network employed zero padding to expand the input and fill in the edges.

This confidence decision model can adaptively select whether to use the trained speech model based on the confidence level of the speech signal. By utilizing a two-channel confidence and decision network, it accurately determines whether the trained speech model should be employed for recognition, considering the input speech features and confidence evaluation. However, it is important to note that the performance of confidence networks heavily relies on the training samples. If the training samples are insufficient or fail to cover various recognition scenarios, the model’s accuracy improvement may be limited. Therefore, when constructing a confidence network, it is crucial to ensure that the training samples are diverse and representative to achieve better performance.

## 4. Analysis of Experimental Results

### 4.1. Acoustic Model Comparison

Speech recognition is a significant research area within the field of artificial intelligence. The most widely used method for speech feature extraction is the mainstream Mel Frequency Cepstral Coefficients (MFCC) method. This method converts speech signals into MFCC coefficient feature vectors, which are then used for speech recognition with the GMM-HMM model. While the MFCC method demonstrates good recognition performance in practical applications, it does have some limitations. For instance, it struggles with processing long-duration sequence data and lacks location sensitivity. In order to address certain limitations of MFCC methods, deep learning techniques based on speech spectrograms have been gradually introduced into the field of speech recognition. The purpose of this experiment is to compare the mainstream MFCC approach with spectral map, DFCNN, and CTC-based approaches. The aim is to identify a model with high accuracy that can serve as a benchmark for subsequent network construction. By conducting the experiments, we aim to determine which method performs better and can provide a solid foundation for further model development. As shown in Table 1, in the experiments conducted in this paper, we selected TensorFlow 1.14 as the primary deep learning framework and utilized it to construct a speech recognition model; we utilized the TIMIT speech dataset, which consists of 6300 pronunciations of American English words. The dataset involves recordings from 630 speakers, with each speaker reading out 10 sentences. Among these sentences, 3000 were randomly selected for the training set, 1000 for the validation set, and another 1000 for the test set. Additionally, we also employed the Chinese speech dataset THCH-30, which is divided into training, validation, and test sets.

For our experiments, we utilized a computer with a specific configuration and installed the necessary software and libraries for TensorFlow 1.14 on it. We also employed a GPU acceleration card to enhance the training and inference efficiency. In order to ensure the reproducibility of the experiments, we meticulously recorded the experimental settings, including hyperparameter values such as learning rate, optimizer type, and number of iterations. Furthermore, we conducted reasonable parameter tuning and cross-validation.

This paper employed the word error rate as a metric to evaluate speech recognition accuracy. The edit distance is calculated by comparing the aligned transcription results with the correct text transcription. A lower word error rate indicates a higher accuracy in the speech recognition system. The same procedure was applied to all speech samples in the test dataset to determine the word error rate for the entire dataset, which was then converted to accuracy. The complete word error rate calculation formula (4) is shown below:(4)$$WER=\frac{\#Deletions+\#Insertions+\#Substitutions}{\#ReferenceWords}$$

The accuracy calculation was performed by iteratively processing all the recorded operations for editing the distance.

In the English dataset TIMIT, the mainstream MFCC method along with the GMM-HMM model was utilized for training and testing. Based on the experimental results, the accuracy of speech recognition using the MFCC method on the test set was 76.5%. In the speech spectrogram, DFCNN, and CTC-based methods, the speech signal was first transformed into a speech spectrogram, and the extracted feature vectors were then inputted into the CTC model for training and recognition. According to the experimental results, the methods based on the speech spectrogram, DFCNN, and CTC achieved an accuracy of 87.2% on the test set, which is 10.7% higher than the mainstream MFCC method.

The performance on the Chinese dataset THCH-30 test set revealed that the accuracy of the MFCC-based method was approximately 80%, while the accuracy of the speech spectrogram, DFCNN, and CTC-based methods surpassed 90%.

Comparing the mainstream speech recognition experiments and the results based on speech spectrograms in Figure 6, (a) for the validation of the results on the dataset Timit; (b) for the validation of the results on the dataset Thch30; the following conclusions can be drawn. The mainstream MFCC methods demonstrate a certain level of recognition ability in speech recognition, but they are weak when it comes to handling long-duration sequence data and location sensitivity. On the other hand, the methods based on speech spectrogram, DFCNN, and CTC exhibit better performance in terms of characterizing speech signals, retaining temporal sequence information, and capturing spectral information compared to the mainstream MFCC methods. As a result, these methods can achieve more accurate speech recognition. Moreover, these methods also exhibit stronger robustness and perform better in the presence of interference factors such as background noise and slurred speech.

### 4.2. ASR System Experimental Results

In order to address challenges such as processing difficulties with domain-specific words, this paper investigated an ASR system based on speech spectrograms and proposes two incremental improvement methods: domain-specific datasets and a confidence model decision based on domain-specific datasets.

One of the methods proposes an approach based on a domain-specific dataset speech network for automatically training domain-specific datasets, resulting in domain-specific speech models. This method utilizes integrated learning to merge the domain-specific speech model with the speech model obtained from training the benchmark dataset. This process automatically enhances the lexical content of the ASR model, improves the adaptability of the language model in new domains or scenarios, and increases the prediction accuracy of the model.

On the other hand, the confidence model decision method based on domain-specific datasets combines a deep fully convolutional neural network with a speech recognition algorithm based on candidate temporal classification, along with a specialized confidence-based classifier. During training, the classifier utilizes the confidence of the dual model as the label input and is capable of predicting not only the sample’s class but also utilizing the accuracy rate as an evaluation metric. This approach effectively enhances the robustness and accuracy of the overall method, enabling the ASR system to select the optimal solution from the dual-channel model for output, thus improving recognition accuracy.

In this paper, a series of experiments were conducted to verify the effectiveness of the proposed method. Two datasets were utilized: a medical symptom speech dataset and the Chinese speech dataset THCHS-30. To demonstrate the effectiveness of each network in the speech recognition system, we adopted a progressive model experimentation approach, where network models were incrementally added. This step-by-step process allowed us to assess the contribution of each network and evaluate the overall performance of the system. Table 2 illustrates the features utilized by different methods on the different datasets.

The results of the specific experimental data, based on the word error rate and character error rate as the evaluation criteria, are as follows:

Experiment 1: A basic ASR system was constructed based on the speech spectrogram, utilizing conventional image-based acoustic features. These features were combined with GMM-HMM models to create a speech recognition system. The experimental results indicate that the speech recognition system performs well in recognizing common words. However, it tends to have false recognitions when encountering uncommon words, resulting in poor performance for domain-specific speech recognition. This limitation is attributed to the lack of specialized domain knowledge in the system’s recognition model;

Experiment 2: To enhance the recognition accuracy of the system, this paper explored the incorporation of domain-specific datasets. Specifically, domain-specific audio datasets are utilized to create a dedicated dataset for the domain. The experimental results demonstrated an improvement in the system’s recognition accuracy upon incorporating domain-specific datasets. This improvement can be attributed to the fact that the domain-specific dataset encompasses information on the specific features of the domain-related speech signals. Consequently, it enables better adaptation to the domain-specific speech dataset, thereby enhancing the accuracy of speech recognition;

Experiment 3: In the aforementioned model, a confidence-based model decision method was introduced. This method enables the selection of the optimal outcome from two candidate results based on the confidence level of the dual-channel model. Convolutional neural networks are employed to construct the confidence-based model, and experimental verification demonstrates significant improvements in the system’s recognition accuracy, highlighting the effectiveness of this approach.

The experimental results, as shown in Table 3 above, indicated that the gradual incorporation of domain-specific datasets and the utilization of a confidence-based model decision method led to significant enhancements in the recognition accuracy of the ASR system for two different datasets. In the first set of experiments, the addition of domain-specific datasets resulted in an approximate 6% improvement in recognition accuracy on the medical dataset test set compared to the baseline system. This indicates a notable enhancement in the performance of the ASR system through the inclusion of domain-specific datasets. In the second set of experiments, the integration of the confidence-based model decision method into the domain-specific dataset system led to an approximately 4% improvement in recognition accuracy on the test set compared to the benchmark system. This highlights the further enhancement in the performance of the ASR system achieved by incorporating the confidence-based model decision method.

To mitigate the impact of random errors on the experimental results, this paper utilized the statistical method of paired t-test to determine if there are significant differences between the sample data. For each of the three systems, ten groups of domain-specific speech samples were randomly selected, and recognition experiments were conducted. The accuracy rates obtained for each system are recorded in Table 4, with labels A representing ASR, B representing ASR+DSL-Net, and C representing ASR+DSL-Net+CD-Net. By employing the paired *t*-test, it is possible to assess whether there exists a statistically significant difference in the performance of these systems on the same sample set.

The significance level was set at 0.05, with v = 9, and the t-boundary table was consulted to obtain $t_{0.05,9}=1.833$. Upon calculating ${Sd}_{A-B}=0.04147,\;t_{1}=2.38,\;t_{1}>t_{0.05,9}$ ${Sd}_{B-C}=0.04613,\;t_{2}=2.41,\;t_{2}>t_{0.05,9}$; These experimental verifications confirmed that the differences were statistically significant, indicating significant variations in the accuracy rates among the three systems.

To further investigate the impact of domain-specific datasets on proprietary domains within this approach, various sizes of loaded feature sets were utilized to validate the model’s recognition effectiveness. As depicted in Figure 7, it is evident that the accuracy of speech recognition gradually improves with an increase in the loading of the feature set. Although the validation set for each iteration is randomized and the dataset may undergo perturbations, resulting in a decrease in accuracy for certain segments, the overall accuracy shows an upward trend. This suggests that the overall efficacy of the method remains unaffected by domain specificity and that the accuracy of speech recognition can be further enhanced by augmenting the size of the domain-specific dataset.

The speech recognition approach proposed in this study incorporates a variety of advanced techniques, such as DSL-Net, CD-Net, DFCNN, CTC, and confidence-based classifiers, with the aim of enhancing the accuracy and adaptability of speech recognition. The method proves particularly advantageous in training and migrating domain-specific datasets, and its effectiveness has been demonstrated in diverse domains when combined with large-scale datasets. This holds significant practical importance for addressing the challenges of speech recognition in real-world application scenarios. 

## 5. Conclusions

The speech recognition approach proposed in this paper is based on domain-specific dataset speech networks and confidence decision networks. The aim of this approach is to enhance existing language models by incorporating external knowledge to better handle unknown words and language rules. In comparison to current approaches in the field of speech recognition, particular attention is given to transformer-based models and listen, attend and spell (LAS) models. These models offer better parallel computation compared to traditional RNN models, utilizing self-attentive mechanisms and positional encoding to capture long-range dependencies. However, the Transformer model may encounter challenges when dealing with long-duration series speech inputs, particularly in the case of large vocabularies and complex speech inputs. In contrast, the method presented in this paper leverages domain-specific datasets and a confidence-based model decision approach optimized for domain-specific speech inputs. As a result, significant accuracy improvements were achieved in processing domain-specific speech recognition tasks. While the LAS model serves as an end-to-end speech recognition model based on an attention mechanism, implementing direct training from acoustic inputs to character level outputs, this paper’s approach introduced domain-specific datasets and a confidence-based model decision method as the primary innovations. The utilization of domain-specific datasets enables more precise training samples, leading to enhanced accuracy and performance of the system. The confidence-based decision method improves recognition accuracy and robustness by selecting candidate results with high confidence levels. In future research, there are plans to further optimize the construction and utilization methods of domain-specific datasets. Additionally, exploring other possible combinations and uses of feature sets will be considered to further enhance the performance and adaptability of the ASR system.

## Figures and Tables

**Figure 1 sensors-23-06036-f001:**
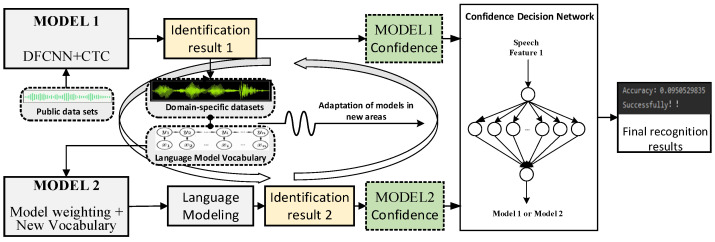
The overall framework of the model.

**Figure 2 sensors-23-06036-f002:**
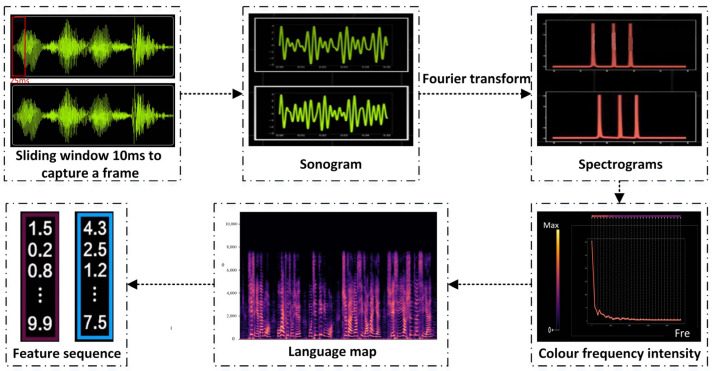
Schematic diagram of feature sequence generation.

**Figure 3 sensors-23-06036-f003:**
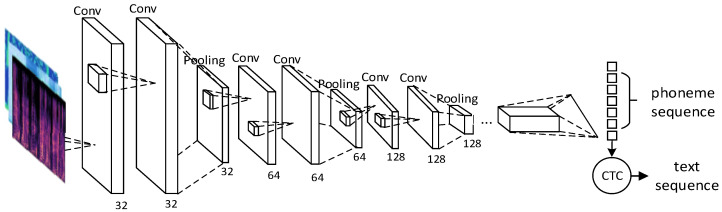
CNN structure diagram.

**Figure 4 sensors-23-06036-f004:**
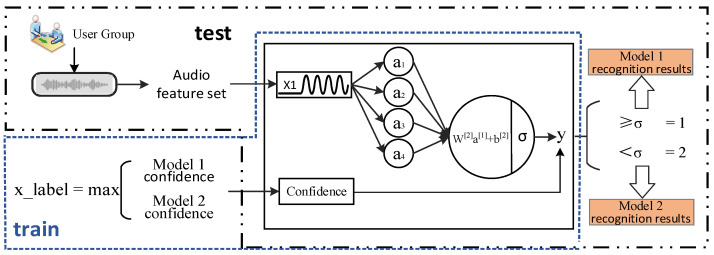
Confidence decision network structure diagram.

**Figure 5 sensors-23-06036-f005:**
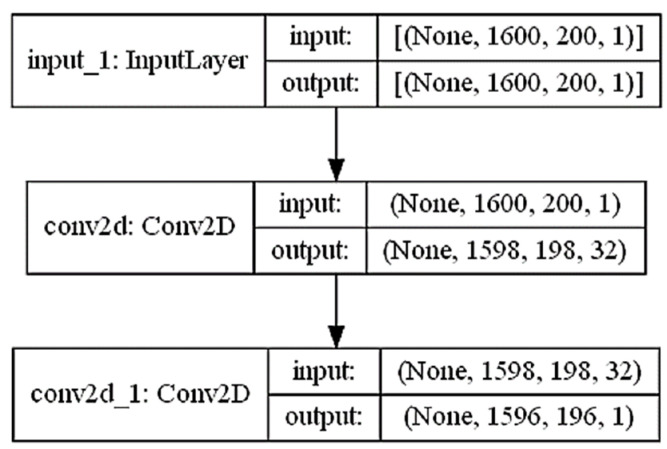
Tensor shape diagram.

**Figure 6 sensors-23-06036-f006:**
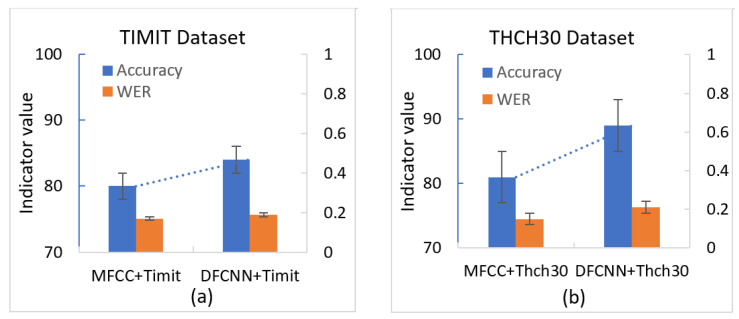
Comparison results of speech recognition experiments.

**Figure 7 sensors-23-06036-f007:**
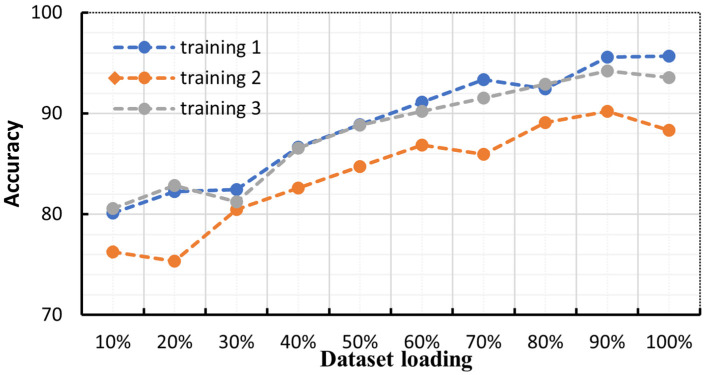
Trend graph of load/correct rate.

**Table 1 sensors-23-06036-t001:** Speech recognition experimental data table.

Method	Dataset	Language Type
DFCNN+CTC	TIMIT	English
MFCC	TIMIT	English
DFCNN+CTC	THCH-30	Chinese
MFCC	THCH-30	Chinese

**Table 2 sensors-23-06036-t002:** Data sheet for model experiments.

Method	Dataset	Characteristics
ASR	Thch-30	Normal
ASR+ DSL-Net	Thch-30	Normal
ASR+ DSL-Net+CD-Net	Thch-30	Normal
ASR	Medical datasets	Medical field
ASR+ DSL-Net	Medical datasets	Medical field
ASR+ DSL-Net+CD-Net	Medical datasets	Medical field

**Table 3 sensors-23-06036-t003:** Comparison table of model experimental results.

Method	Dataset	Accuracy	S	D	I
ASR	Thch-30	85%	4.12	0.98	0.74
ASR+ DSL-Net	Thch-30	91%	3.27	0.88	0.69
ASR+ DSL-Net+CD-Net	Thch-30	95%	2.59	0.46	0.49
ASR	Medica	82%	5.08	1.21	1.04
ASR+ DSL-Net	Medica	87%	3.91	0.92	0.87
ASR+ DSL-Net+CD-Net	Medica	91%	3.21	0.54	0.71

**Table 4 sensors-23-06036-t004:** Paired sample t-test results.

	1	2	3	4	5	6	7	8	9	10
**A**	0.157	0.163	0.153	0.127	0.102	0.183	0.152	0.098	0.119	0.087
**B**	0.128	0.177	0.11	0.121	0.139	0.113	0.073	0.121	0.143	0.092
**C**	0.081	0.092	0.128	0.083	0.085	0.118	0.063	0.094	0.073	0.107
**d** **A-B**	0.029	0.014	0.043	0.006	0.037	0.07	0.079	0.023	0.024	0.005
**d** **B-C**	0.047	0.085	0.018	0.038	0.054	0.005	0.01	0.027	0.07	0.015

## Data Availability

Not applicable.

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
