# Peer review of "A Speech Recognition Method Based on Domain-Specific Datasets and Confidence Decision Networks"

_sensors, 2023, doi:10.3390/s23136036_

Round 1
Reviewer 1 Report
A Speech Recognition Method Based on Domain-Specific
Datasets and Confidence Decision Networks
-----------------
The paper proposes speech recognition methods using domain-specific datasets and confidence decision networks.
The paper is interesting to read and follow; however, it requires the following issues to be fixed for consideration.
1. It is unclear why this approach is required. Please succinctly mention this in the abstract.
2. Quantitative summary of the findings should be presented in the abstract.
3. The paper needs to discuss recent SOTA audio representation techniques. The popular approaches have been discussed
in the following papers:
https://ieeexplore.ieee.org/abstract/document/9784899/
Please discuss them in the introduction part of the paper as they detail the recent audio representation techniques.
4. The paper requires comparing the proposed method with the SOTA methods
5. Please put statistical significance tests in the paper, such as paired t-test/Wilcoxon ranked test.
Reviewer 2 Report
The authors propose an acoustic modeling approach that combines the speech spectrogram, DFCNN, and CTC, aiming to achieve improved speech recognition results. This approach effectively utilizes the rich information provided by the speech spectrogram, the powerful feature extraction capability of DFCNN, and the sequence modeling capabilities of CTC without alignment. By accommodating speech signals of varying lengths, the proposed approach shows promise in enhancing speech recognition performance.
Additionally, the authors address the challenge of unfamiliar words in new domains by presenting a comprehensive system that incorporates N-gram technology and the construction of domain-specific datasets into language models. This innovative approach enables speech recognition in new domains, expanding the applicability of the proposed system.
To further optimize the model, the authors propose a speech confidence-based determination method. This method dynamically adjusts the use of the language model, effectively enhancing the accuracy of the speech recognition model. By incorporating confidence-based decision-making, the proposed method demonstrates potential for improving the overall performance of the speech recognition system.
Overall, the proposed approach combines multiple techniques and methodologies to tackle various challenges in speech recognition, including handling speech signals of varying lengths, addressing unfamiliar words in new domains, and optimizing model performance through confidence-based determination. The findings presented in this study have the potential to contribute to the advancement of speech recognition technology and can be of interest to researchers and practitioners in the field.
Room for improvements:
1. The authors have presented the proposed method very well. However, I did not find enough details on the experimental setup, apart from the information about the datasets used. It would be helpful to include information on the testbed or software used to simulate the experiment.
2. The authors need to explicitly mention the research goal for their experimental setup. It is important to clearly state the specific objectives or goals that the experiments aim to achieve.
3. It is unclear whether the authors implemented their models from scratch or built upon existing models. It would be beneficial to provide clarification on whether the proposed method is entirely novel or if it builds upon or enhances any pre-existing models.
4. The process of obtaining the accuracy results is not described. Did the authors write any code and implement it? It would be helpful to explain the methodology used to measure accuracy and provide details on any code implementation, if applicable.
5. The limitations and the validity of the proposed method are missing. It is important to address the limitations or potential drawbacks of the proposed approach and discuss the validity of the results obtained. This would provide a more comprehensive understanding of the strengths and weaknesses of the method.
In Tables 1, 2, and Figure 6, since the authors are comparing the results, it is essential to explicitly mention whether they are comparing with previous models or their own models. If comparing with previous studies, citations are required to establish the context and provide a basis for comparison. Therefore, it is crucial to define the research goal(s) before setting up an experiment to ensure clarity and proper evaluation of the proposed method.
Reviewer 3 Report
This paper proposed an automatic speech recognition (ASR) model, based on domain-specific data selection for adaptation and model selection based on confidence score. For adaptation, CNN-based architecture was used with CTC loss function. In addition, dynamic lexicon expansion was considered for adaptation. The experiment results demonstrate that the proposed methods can reduce ASR systems' word error rate (WER) on the target domain.
In this paper, the authors mentioned that to alleviate the limitations of spectrogram and CTC-based acoustic model (AM) training, they proposed spectrogram, DFCNN, and CTC-based AM training architecture. Several of the mentioned reference papers use the deep CNN architecture for AM training. Extraction of spectrograms is a known method in the speech community. This is better to clarify the advantages of the proposed method w.r.t. the reference papers. In addition, recently several papers are using attention-based architectures for handling long-time sequences. We can not consider the CNN-based architectures as SOTA methods.
The overall model framework (Figure 1) needs more clarification. Based on this figure, MODEL 1 was trained using public datasets which shows a contradiction with the explanation in section 3.1. In section 3.3, this is not mentioned how the weight coefficient a1 and a2 in question 3 are set. In the experiment section, this is not explained how the accuracy is computed in Figure 6. Some of the references are not in the standard format e.g., ref. [22], and the first author of this paper is not Ehsan as mentioned in the related work.
There are several typos in the paper including line 129 "vour paper" -> "our paper", Figure 4 x_lable -> x_label.
Round 2
Reviewer 1 Report
Thanks to the authors for their hard work to improve the quality of the manuscript. The manuscript is acceptable now.
Author Response
Dear reviewer:
Thank you very much for your diligent effort in reviewing our manuscript. We truly appreciate your hard work and dedication. We are glad to hear that you find the manuscript now acceptable. Your valuable feedback and suggestions are crucial to us, and they have played a significant role in improving our research work.Once again, we extend our sincere gratitude for your contribution and professionalism throughout the review process.
Thank you once again for your invaluable support!
Best wishes,,
Zhe Dong, Qianqian Ding, Weifeng Zhai, Meng Zhou.
Reviewer 2 Report
The authors addressed all the questions raised in my first review. Please proofread the paper thoroughly before it get published.
Author Response
Dear reviewer:
We would like to express our sincere appreciation for your diligent efforts in reviewing our manuscript and providing valuable feedback. We also acknowledge your recommendation to proofread the manuscript thoroughly before its publication. We assure you that we will pay utmost attention to this aspect and ensure that the final version of the paper is meticulously reviewed for any errors or inconsistencies.
Thank you sincerely for your valuable contributions.
Best wishes,,
Zhe Dong, Qianqian Ding, Weifeng Zhai, Meng Zhou.
Reviewer 3 Report
The requested modifications were covered in this version. I suggest comparing the proposed method with the transformer-based ASR systems. In section 4.1 to clarify the process of accuracy computation this is better to define an equation rather than showing the Python code.
Some hyphens must be removed in the conclusion section.
